# Efficacy of Penile Low-Intensity Shockwave Therapy and Determinants of Treatment Response in Taiwanese Patients with Erectile Dysfunction

**DOI:** 10.3390/biomedicines9111670

**Published:** 2021-11-12

**Authors:** Kai-Yi Tzou, Su-Wei Hu, Oluwaseun Adebayo Bamodu, Yuan-Hung Wang, Wen-Ling Wu, Chia-Chang Wu

**Affiliations:** 1Department of Urology, School of Medicine, College of Medicine, Taipei Medical University, Taipei 110, Taiwan; 11579@s.tmu.edu.tw; 2Department of Urology, Shuang Ho Hospital, Taipei Medical University, New Taipei City 235, Taiwan; 10352@s.tmu.edu.tw (S.-W.H.); 16625@s.tmu.edu.tw (O.A.B.); 15334@s.tmu.edu.tw (W.-L.W.); 3TMU Research Center of Urology and Kidney (TMU-RCUK), Taipei Medical University, Taipei 110, Taiwan; 4Graduate Institute of Clinical Medicine, College of Medicine, Taipei Medical University, Taipei 110, Taiwan; d508091002@tmu.edu.tw; 5Division of Hematology and Oncology, Department of Internal Medicine, Shuang Ho Hospital, Taipei Medical University, New Taipei City 235, Taiwan; 6Department of Medical Research, Shuang Ho Hospital, Taipei Medical University, New Taipei City 235, Taiwan

**Keywords:** erectile dysfunction, extracorporeal shockwave therapy, Li-ESWT, IIEF-5, EHS, minimal clinically important difference, MCID, independent predictors, treatment success, ethnogeography

## Abstract

Background: Erectile dysfunction (ED) remains an emotional wrench to patients and a therapeutic challenge to urologists in andrology clinics worldwide. This is, in part, related to refraction to, or transient effect of phosphodiesterase 5 inhibitors (PDE5i), coupled with patients’ dissatisfaction with this treatment modality. Low-intensity extracorporeal shockwave therapy (Li-ESWT) is an evolving treatment option, with promising curative potential. Current international guidelines are inconclusive, bear weak recommendation strength, and lack ethnogeographic consensus. Objectives: This study evaluated the safety, efficacy, and effect duration of Li-ESWT, as well as exploring disease-associated determinants of treatment success in Taiwanese males with ED. Methods: A cohort of 69 eligible cases treated with 12 sessions of Li-ESWT and followed up for at least 12 months after treatment, between January 2018 and December 2019 at our medical facility, was used. The present single-center, retrospective, non-randomized, single-arm study employed standardized erectile function evaluation indices, namely, the five-item International Index of Erectile Function (IIEF-5) and Erection Hardness Score (EHS). Clinicopathological analyses of selected variables and comparative analyses of time-phased changes in the EF indices relative to baseline values were performed. Evaluation of treatment success was based on minimal clinically important difference (MCID), using a binomial logistic regression model. Results: The median age and duration of ED for our Taiwanese cohort were 55 years and 12 months, respectively, and an average of 31.3% presented with co-morbidities. The mean improvement in IIEF-5, EHS, and quality of life (QoL) domain scores relative to the baseline values was statistically very significant (*p* < 0.001) at all indicated follow-up time-points. When stratified, Taiwanese patients with severe and moderate ED benefited more from Li-ESWT, compared with those in the mild or mild-to-moderate group. Patients’ pre-Li-ESWT PDE5i response status was not found to significantly influence Li-ESWT response. Univariate analysis showed that age > 45 years (*p* = 0.04), uncontrolled diabetes mellitus (*p* = 0.04), and uncontrolled hyperlipidemia (*p* = 0.01) were strongly associated with Li-ESWT efficacy; however, only age > 45 years (*p* = 0.04) and uncontrolled hyperlipidemia (*p* = 0.03) were found to be independent negative predictors of Li-ESWT success by the multivariate logistic model. Follow-up was uneventful, with no treatment-related adverse events or side effects reported. Of the treated patients, 86.1% indicated satisfaction with the treatment regimen, and over 90% indicated they would recommend the same therapy to others. Conclusions: Li-ESWT is a safe and efficacious therapeutic modality for Taiwanese patients with ED. Uncontrolled hyperlipidemia and age > 45 years are independent negative predictors of treatment success for this cohort.

## 1. Introduction

Erectile dysfunction (ED), entailing the recurrent or consistent inability to achieve and/or maintain penile erection sufficient for coitus, remains a clinically challenging male-specific medical condition with high incidence and quality of life (QoL) implications, globally, especially with one in every two males over the age of 40 years being affected [1,2]. Aging and the presence of co-morbidities such as diabetes mellitus (DM), hypertension (HTN), hyperlipidemia, hypogonadism, cardiovascular diseases (CVDs), and tobacco smoking have been implicated in the high incidence and severity of this multifactorial pathology [1,3].

Despite improved understanding of the pathoetiology of ED, advances in diagnostics, and novel therapeutic strategies, ED continues to defy treatment success, with high prevalence. In fact, following the advent of phosphodiesterase 5 inhibitors (PDE5i) in the late 1990s, and their adoption as first-line therapy for ED [2], the touted success of PDE5i has been beleaguered by suboptimal satisfaction from about 50% of all treated patients [4]. This disparaging therapy satisfaction status, which in part is attributable to disease duration and severity, has been suggested to affect patients’ compliance and the perceived efficacy of PDE5i [4,5], thus necessitating therapy recalibration to elicit optimal satisfaction or initiation of an alternative highly efficacious therapeutic modality in patients with ED that are non-responsive or poorly responsive to PDE5i.

Dating back to the late 1970s, low-intensity shockwave therapy (Li-ESWT) has been widely used in various fields of medicine, including urology, gastroenterology, orthopedics, and cardiovascular medicine [6]. Following the seminal work of Vardi et al. demonstrating the applicability of Li-ESWT in andrology [7,8], there is accruing evidence of the therapeutic benefit of Li-ESWT in patients with ED [8,9,10]. Previous studies have associated the observed therapeutic benefit with the intrinsic ability of Li-ESWT to up-regulate growth factors, promote endothelial progenitor cell proliferation and activation, induce neovascularization, increase the production of nitric oxide, elicit tissue reperfusion and regeneration, and consequently improve erectile function [11,12,13].

Regardless of the reported safety and tolerability of Li-ESWT in patients with ED, the European Society of Sexual Medicine (ESSM) has served a note of caution, alluding that “its efficacy for the treatment of ED is doubtful and deserves more investigation” [13]. Moreover, the European Association of Urology (EAU) guidelines on ED indicate that the use of Li-ESWT is permissible for patients with mild organic ED or poor responders to PDE5i, but with a weak recommendation strength, while the American Association of Urology (AUA) guidelines on ED, finding insufficient evidence for or against the modality, make a conditional recommendation of grade C that Li-ESWT should be considered investigational [14]. This cloud of controversy around the use of Li-ESWT coupled with reported patient dissatisfaction or perceived inefficacy informs the present study in our Taiwanese cohort.

The use of Li-ESWT for patients with ED was initiated in our institution in 2018. Thus, the present study evaluates the therapeutic efficacy and safety of Li-ESWT, as well as exploring factors that mitigate the efficacy of Li-ESWT or negative predictors of therapy response in Taiwanese patients with ED. Therapy response and clinical outcome were assessed using established indices of erectile function, namely, International Index of Erectile Function (IIEF), a patient-reported measure of erectile dysfunction and other sexual issues [15], and Erection Hardness Score (EHS), a one-item scale with validated psychometric attributes for accurate assessment of EF in PDE5i-treated patients with ED [16], coupled with the minimally important clinical difference (MICD), which is “the smallest measurable change in outcome score that will make a difference to the patient” [17,18]. Consistent with data from Caucasian cohorts, the data presented herein demonstrate that Li-ESWT is an efficacious and safe treatment modality for most of our Taiwanese patients with ED, with 86.1% of treated patients indicating satisfaction with the treatment regimen, and over 90% indicating that they would recommend the same therapy to others. Nonetheless, this study also found certain ethnogeography-specific differences and determinants of treatment success or failure in Taiwanese patients, compared with other ethnic groups.

## 2. Methods

### 2.1. Study Design and Patients

The present single-center, retrospective, non-randomized, single-arm study, approved by the Joint Institutional Review Board of Taipei Medical University (Approval No.: N20202005032), was compliant with the Declaration of Helsinki guidelines on studies involving human subjects, and the need for written informed consent was waived because of the retrospective nature of the study. The study enrolled 85 non-psychogenic cases who were treated with Li-ESWT between January 2018 and December 2019, at the Department of Urology, Shuang Ho Hospital, Taipei Medical University, New Taipei, Taiwan. All patients, regardless of PDE5i response status, underwent the same treatment protocol. All sexual and medical history, physical examination, and laboratory workup results were respectively retrieved from the institutional electronic medical archive, including patients’ age, anthropometric data, body mass index (BMI), severity and duration of ED, blood testosterone level, and history of PDE5i use.

### 2.2. Inclusion and Exclusion Criteria

A case was included if the patient was aged ≥ 18 years, with definitive non-psychogenic ED diagnosis, had an ED history ≥ 6 months, non-responsive to or dissatisfied with administered medical therapies including PDE5i for at least the last 6 months, had an EHS ≤ 2, was already off PDE5i in the last 3 weeks before Li-ESWT initiation, and completed 12 sessions of Li-ESWT and followed up to one year, while maintaining regular sexual activity. Cases were excluded from the study if ED was due to a post-radical prostatectomy, post-radiation therapy for pelvic organs, or post-chemotherapy pathoetiology, and untreated hypogonadism. Cases with documented anatomical, hormonal, or neurological impairments, or that were concomitantly receiving treatment for any psychiatric condition, were also excluded. Ultimately, 16 of the enrolled 85 cases were excluded, leaving 69 eligible cases. ED was diagnosed based on initial symptomatology self-reported by patients, and etiology confirmed through completed EF questionnaires, complete medical workup, and sexual history. Psychogenicity was ruled out when etiology was uncertain or unknown; however, in the presence of psychogenic factors, including hypoactive sexual desire, major psychiatric disorders such as anxiety disorder and depression, pessimistic attitudes, negative outlook on life, relationship problems, or stress, the condition was adjudged psychogenic, and thus excluded [15,16,17,18,19,20].

### 2.3. Li-ESWT Protocol

All patients were treated with the DUOLITH^®^ SD1 mobile shockwave therapy apparatus with a SEPIA^®^ handpiece (Storz Medical AG, Tägerwilen, Switzerland). Each 20-min treatment session performed once per week for 12 consecutive weeks was carried out in the outpatient setting. During each treatment session, a total of 3000 shockwave impulses at an energy flux density of 0.2 mJoule/mm^2^ and a frequency of 4 Hertz were given, with 500 impulses applied to six points on the penis, namely, the bilateral *crura* (proximal projections of the *corpora cavernosa*), 2 and 10 o’clock direction for the penile shaft (aiming at the *corpora cavernosa*), and distal penile shaft (just inferior to the *glans penis*). All procedures in the study were performed by a single highly skilled urologist (Dr. Chia-Chang Wu). For the duration of the Li-ESWT, patients maintained their regular sexual activity.

### 2.4. Assessment of Li-ESWT Efficacy Using Erectile Function (EF) Indices

Response to Li-ESWT and clinical outcome were assessed using established indices of erectile function, namely, International Index of Erectile Function (IIEF), a patient-reported measure of erectile dysfunction and other sexual issues [15], and Erection Hardness Score (EHS), a one-item scale with validated psychometric attributes for accurate assessment of EF in PDE5i-treated patients with ED [16], coupled with the minimally important clinical difference (MICD) or minimal clinically important difference (MCID), which is “the smallest measurable change in outcome score that will make a difference to the patient” [17,18]. EF assessments were performed at baseline (pre-treatment), and 1 month, 3 months, 6 months, and 12 months post-treatment.

### 2.5. Li-ESWT Outcome Measurement

Li-ESWT success was defined primarily based on the MCID, with an improvement ≥ 7, 5, or 2 points for patients with baseline IIEF-5 scores of 5–7 (severe ED), 8–16 (moderate ED), or 17–21 (mild ED) points, respectively. Secondarily, treatment success was adjudged when erection was hard enough for vaginal penetration (EHS ≥ 3).

### 2.6. Statistical Analysis

Binomial logistic regression with univariate and multivariate models was carried out to delineate determinants of Li-ESWT success or negative predictors of therapy response. Pearson’s chi-squared (χ2) test was used to determine the relationship or association between categorical variables. Normality of continuous data distribution was determined by the Shapiro–Wilk test and normal Q-Q plots. The paired *t*-test was used for comparing continuous data. The Student’s *t*-test was used to assess improvement in erectile function based on IIEF-5, quality of life (QoL), and EHS scores. *p*-Values ≤ 0.05 were considered statistically significant. All statistical analyses were performed using IBM SPSS Statistics for Windows, Version 25.0 (IBM Corp. Released 2017, Armonk, NY, USA: IBM Corp).

## 3. Results

### 3.1. Taiwanese Patients with ED Are Mostly Older Than 45 Years and Present with Advanced Disease Severity

A total of 69 patients with non-psychogenic ED, who received Li-ESWT at our institution between 2018 and 2019, were eligible, according to the inclusion and exclusion criteria. As shown in Table 1, though the median age was 55 years (range: 32–77), 73.91% of the cohort were older than 45 years, with a median BMI of 24.6 ± 3.89 kg/m^2^, and median baseline IIEF-5 or EHS score of 10 ± 5.54 or 2.0 ± 0.80, respectively. The median duration of ED was 12 ± 26.03 months, with moderate and severe cases constituting 63.77% of the whole cohort, while 8.6% were mild cases and 23.2% were mild-to-moderate cases (Table 1). Consistent with reports that indicate an association between ED and other morbidities [21], we found an average of 31.3% patients with co-morbidities, namely, DM, HTN, hyperlipidemia, chronic kidney disease (CKD), and hypogonadism (Table 1). Moreover, of the eligible cases, 52 (75.4%) were non-respondent to PDE5i (Table 1).

### 3.2. Li-ESWT Significantly Improves the Erectile Function of Taiwanese Patients Compared with Baseline

To evaluate the therapeutic efficacy of Li-ESWT, we employed known EF indices over the duration of follow-up. We observed that the mean IIEF-5 score of the total cohort increased from 10.33 ± 5.43 at baseline to 16.98 ± 5.82 at 1-month follow-up (6.65 point increase, *p* < 0.001). Similarly, 6.37 (*p* < 0.001), 5.79 (*p* < 0.001), and 5.10 (*p* < 0.001) point improvements in the mean IIEF-5 score were observed at the 3-month, 6-month, and 12-month follow-up time-points, respectively (Table 2). Akin to the IIEF-5 scores, we found that compared with the baseline EHS of 2.1 ± 0.8, the cohort mean EHS increased by 0.80–0.89 points (*p* < 0.001) over the course of follow-up (Table 2). Since contextualizing therapy-induced changes in terms of clinically relevant improvement is fundamental for an objective evaluation of treatment efficacy, proper interpretation of results across studies, and effective management of patients, adopting Rosen et al.’s MCID model for treatment success, we found that Li-ESWT elicited a clinically relevant improvement in IIEF-5 for 56.52%, 53.62%, 55.07%, and 46.38% of patients with ED at the 1-month, 3-month, 6-month, and 12-month follow-up time-points, respectively.

### 3.3. Taiwanese Patients with Advanced ED Severity Benefit More from the Therapeutic Effect of Li-ESWT, Compared with the Less Severe Cases

Because of the relevance of patient stratification in therapy response and clinical outcome, observing in Table 1 that the moderate and severe cases constituted ~64% of the whole cohort, while ~9% and ~23% were mild and mild-to-moderate cases, respectively, we next evaluated the therapeutic effect of Li-ESWT according to ED severity class. We also found that compared with the mild and mild-to-moderate cases, Taiwanese patients with moderate and severe ED exhibited significantly greater improvement in the mean EF indices all through the follow-up duration, as demonstrated by increases of 9.17 (*p* < 0.001), 6.50 (*p* < 0.001), 4.94 (*p* < 0.001), and 1.67 (*p* < 0.05) in the relative mean IIEF-5 of patients with severe, moderate, mild-to-moderate, and mild ED, respectively, at 1-month follow-up (Figure 1A, also see Appendix A). A similar trend was observed at the 3-month follow-up (severe: 8.54 (*p* < 0.001), moderate: 6.35 (*p* < 0.001) vs. mild-to-moderate: 5.31 (*p* < 0.001), mild: 0.89), 6-month follow-up (severe: 7.80 (*p* < 0.001), moderate: 5.80 (*p* < 0.001), vs. mild-to-moderate: 4.56 (*p* < 0.001), mild: 1.00), and 12-month follow-up (severe: 7.00 (*p* < 0.001), moderate: 4.80 (*p* < 0.001), vs. mild-to-moderate: 4.19 (*p* < 0.01), mild: 0.33) (Figure 1A, also see Appendix A). Concomitantly, we also observed that compared with their peers, patients with severe ED achieved the greatest increase in EHS at all follow-up time-points (*p* < 0.01) (Figure 1B, also see Appendix A). Moreover, our MCID-based assessment of therapy response/success showed that the Li-ESWT was most beneficial to patients with mild-to-moderate and severe ED, compared with the mild and moderate cases, at all the follow-up time-points (Figure 1C, also see Appendix A). These data indicate, at least in part, that Li-ESWT prevents symptom exacerbation and alleviates the severity of ED in Taiwanese patients.

### 3.4. Li-ESWT Is Therapeutically Efficacious in Taiwanese Patients Regardless of Their PDE5i Response Status, Albeit Slightly More So among PDE5i Responders

Against the background of the EAU guidelines permitting the use of Li-ESWT for patients with mild organic ED or poor responders to PDE5i, we further comparatively analyzed the therapeutic effect of Li-ESWT in Taiwanese patients with ED who were responsive or non-responsive to PDE5i. As shown in Appendix A, Li-ESWT was effective in both PDE5i response subgroups. MCID in IIEF-5 score was achieved in 58.8%, 58.8%, 47.1%, and 47.1% of PDE5i responders versus 55.7%, 51.9%, 57.7%, and 46% of the PDE5i non-responders, at 1-month, 3-month, 6-month, and 12-month post-therapy follow-up, respectively (Figure 2A–C, also see Appendix A). The inter-group differences in Li-ESWT success were statistically non-significant (*p* > 0.05) across all follow-up time-points. Consistent with the MCID, we also found that the mean IIEF-5 improvement was slightly less in the PDE5i non-responders, albeit statistically insignificantly (0.40–0.83, *p* > 0.05) across all follow-up time-points, except for 12-month follow-up, where it was 0.23 points greater than that achieved by the PDE5i responders (*p* = 0.89) (Figure 2A,C, also see Appendix A). A similar trend was observed for the QoL domain of the IIEF-5, which was ambivalent for both PDE5i response groups, except for the 12-month follow-up where the PDE5i non-responders enjoyed a 1.3 points QoL advantage (*p* = 0.56) (Figure 2D). We also observed that the mean EHS increase for the PDE5i non-responders was 1.31-fold (*p* = 0.36), 1.50-fold (*p* = 0.17), 1.44-fold (*p* = 0.23), and 1.91-fold (*p* = 0.08) greater than for the responders at 1-month, 3-month, 6-month, and 12-month follow-up (Figure 2B, also see Appendix A).

### 3.5. Age > 45 Years and Uncontrolled Hyperlipidemia Are Independent Negative Predictors of Li-ESWT Response or Success in Taiwanese Patients with ED

Having shown that Taiwanese patients with severe and moderate ED benefit more from the therapeutic effect of Li-ESWT, compared with their counterparts with mild and mild-to-moderate, and that there is no stringent dependency of Li-ESWT therapeutic efficacy on PDE5i response status, we further explored factors that may affect response to Li-ESWT in Taiwanese patients. Univariate analysis revealed that age > 45 years (OR = 0.27, *p* = 0.04), uncontrolled DM (OR = 0.17, *p* = 0.04), and uncontrolled hyperlipidemia (OR = 0.25, *p* = 0.01) were associated with reduced or non-response to Li-ESWT (Table 3). Consistent with the univariate analysis results, age > 45 years (odds ratio, OR = 0.24, *p* = 0.04) and uncontrolled hyperlipidemia (OR = 0.27, *p* = 0.03) were found to be independent risk factors for Li-ESWT failure; however, uncontrolled DM (OR = 0.21, *p* = 0.08) was not an independent risk factor in this model (Table 3). As shown in Table 3, univariate and multivariate analyses revealed that tobacco smoking, hypogonadism, HTN, DM, hyperlipidemia, duration of ED > 2 years, EHS < 3, and non-response to PDE5i had no negative influence on the response to or treatment success of Li-ESWT for Taiwanese patients with ED.

## 4. Discussion

Secondary to circulatory, psychogenic, hormonal, neurogenic, and/or pharmacological factors, erectile dysfunction (ED) is a multifactorial incapacity to achieve or maintain penile erection that is satisfactory for coitus. Despite substantial diagnostic and therapeutic advances, coupled with our increased understanding of the pathoetiology and biology of ED, the prevalence of ED remains unabated, as incidence continues to soar with age.

A recent report indicates that up to 76% of treated patients with ED opt to discontinue their treatment with PDE5i inhibitors secondary to several reasons, including perceived and/or objective treatment ineffectiveness, drug-related adverse events/side effects, unsatisfactory quality of coitus, and cost of treatment [22].

Mindful of patients’ dissatisfaction or subjectively perceived inefficacy of first-line PDE5i [22], as well as the inconclusive global consensus, conditional recommendations, and weak recommendation strength of guidelines for use of Li-ESWT in managing ED [13,14], the present study showed that the Taiwanese patients with ED who participated in this study were mostly older than 45 years and presented with moderate and severe ED. This observation aligns with our perception of the ethnogeographic variability in health experiences, susceptibility to disease, and/or therapy response. Moreover, this finding is in part consistent with the contemporary understanding that “ED is one of the most common conditions affecting middle-aged and older men” [23], especially with findings from the Massachusetts Male Aging Study (MMAS) indicating that more than one in every two males aged 40–70 years old presents with some form of ED [19,23]. Moreover, our finding that Taiwanese patients with ED present with other concomitant morbidities, including DM, HTN, hyperlipidemia, CKD, and hypogonadism, aligns with findings from various longitudinal and cross-sectional studies, linking ED with depression, metabolic syndrome, hyperlipidemia, DM, HTN, lower urinary tract symptoms (LUTS), and CVDs [1,2,21].

It is clinically relevant that while on a generic note, Li-ESWT significantly improves the erectile function of Taiwanese patients compared with baseline, upon severity-based stratification, patients with severe and moderate ED, and not the mild and mild-to-moderate cases, benefited most from the therapeutic effect of Li-ESWT. This is in contrast to the non-consensual findings of randomized controlled trials indicating that “Li-ESWT may be more beneficial in cases of mild ED or when combined with PDE5 inhibitors in men with moderate to severe ED”, obtained from mainly Caucasian cohorts [24]. Moreover, the greater therapeutic success of Li-ESWT in these advanced severity cases in our Taiwanese cohort, contradicts the recommendation of the EAU guidelines on ED indicating that the use of Li-ESWT is permissible for patients with mild organic ED [14]. While we cannot fully explain this contradiction, we believe this divergence is consistent with our evolving perception of the ethnogeographic variability of health experiences, susceptibility to disease, and/or therapy response. Based on our data, we posit that, at least in part, Li-ESWT prevents symptom exacerbation, and alleviates the severity of ED in Taiwanese patients.

In addition, we demonstrated that Li-ESWT is therapeutically efficacious in Taiwanese patients regardless of their PDE5i response status, albeit slightly more so among PDE5i responders. Our findings are concordant with results reported by Spivak Leonid’s team indicating that Li-ESWT is an effective and safe therapeutic modality for PDE5i responsive or non-responsive patients with ED, despite finding that the PDE5i responders outperformed the PDE5i non-responders, and that this was statistically significant [25]. This is corroborated by results from a double-blind, placebo-controlled study demonstrating that Indian men with vasculogenic ED who were PDE5i responders enjoyed long-term improvement in their EF following Li-ESWT, unlike the PDE5i non-responders [26].

Furthermore, we found that age > 45 years and uncontrolled hyperlipidemia are independent negative predictors of Li-ESWT response/success for Taiwanese patients with ED. This is partially consistent with the report of Hisasue et al. that age and presence of co-morbidities are negative predictors of response to Li-ESWT [21]. However, with 71% aged less than 65 years, we did observe that our Taiwanese patients with ED were relatively younger than in several published results from Caucasian cohorts [19,21,23]. Most studies found that the course, severity, and/or treatment response of ED are associated with certain co-morbidities, namely, HTN, DM, hyperlipidemia, and CVDs [1,21,27,28]. In contrast, our present study found that neither HTN, DM, nor hyper-lipidemia was associated with ED. We found rather that uncontrolled DM and uncontrolled hyperlipidemia were associated with ED, while only uncontrolled hyperlipidemia was identified as an independent co-morbid negative predictor of response to Li-ESWT for Taiwanese patients with ED. Moreover, while this finding is partially incongruous with those of Vita et al., wherein age ≥ 65 years, DM, and hypercholesterolemia were associated with early non-responsiveness or reduced response to Li-ESWT [29], it is clinically relevant in terms of patient stratification for personalized medicine and effective management of ED cases in Taiwan, taking into account patients’ individual variability in disease susceptibility and therapy response, while drawing out management plans [30].

As demonstrated in the present study, for Li-ESWT success in Taiwanese ED cases, uncontrolled hyperlipidemia, and not just a history of hyperlipidemia, is a critical response factor that must be addressed. Consistent with our evolving understanding of ED as essentially a pathology of endothelial dysfunction [31,32,33,34], it is translationally rational that uncontrolled hyperlipidemia may contribute to ED by promoting endothelial dysfunction [35], and this highlights a probable role for complementary antilipidemics, such as statins and fibrates, in potentiating Li-ESWT therapeutic effect and alleviating ED severity by improving endothelial function, as part of their pleiotropic pharmacological activities [20,36,37] in patient subgroups where Li-ESWT as a sole therapy has under-performed.

In terms of EF indices, it is notable that while the reported improvement in the mean total IIEF-5 score, relative to baseline, at 3-month post-Li-ESWT follow-up ranged from 2.2 to 2.8 points [29,38,39], an improvement of over 6 points was achieved in our Taiwanese cohort, at the same time-point. However, in contrast to the relatively high treatment success rates of 62%, 74%, and 83% after 6, 12, and 18 sessions reported by Dimitrios Kalyvianakis’s team [15], 12 sessions of Li-ESWT elicited a subpar success rate of 59.6% among Taiwanese patients with ED.

### 4.1. Limitations

As with any study of this nature, the present study has some limitations. First, the use of a relatively small sample size may be suggestive of higher variability and lower reliability of the findings reported herein. Secondly, the single-arm study design may make it difficult, if not impossible, to rule out “placebo effect” or “spontaneous resolution” and to properly characterize the actual effect of Li-ESWT as a treatment modality. While longer than the duration in many published works, we still consider our 1-year follow-up duration after treatment to be short-term, as this did not allow for proper validation of the transient or permanent nature of the Li-ESWT-elicited therapeutic effect.

### 4.2. Conclusions

In conclusion, the data presented in this study demonstrate the efficacy, tolerability, and safety of Li-ESWT as a therapeutic modality for Taiwanese patients with ED, with uncontrolled hyperlipidemia and age > 45 years identified as independent negative predictors of treatment success, while highlighting conflicting findings with a probable ethnogeographic connotation.

## Figures and Tables

**Figure 1 biomedicines-09-01670-f001:**
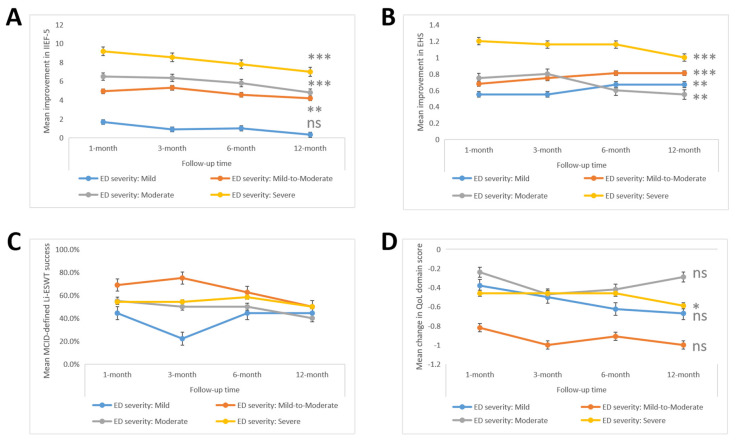
The therapeutic efficacy of Li-ESWT in patients stratified by ED severity. Line graphs showing the (**A**) mean improvement in IIEF-5, (**B**) mean improvement in EHS, (**C**) mean MCID-defined Li-ESWT success, and (**D**) mean change in QoL domain score in patients with mild, mild-to-moderate, moderate, or severe ED, at indicated time-points. ns, non-significant; * *p* < 0.05, ** *p* < 0.01, *** *p* < 0.001; Li-ESWT, low-intensity extracorporeal shockwave therapy; ED, erectile dysfunction; IIEF-5, five-item International Index for Erectile Function; EHS, Erection Hardness Score; MCID, minimal clinically important difference; QoL, quality of life.

**Figure 2 biomedicines-09-01670-f002:**
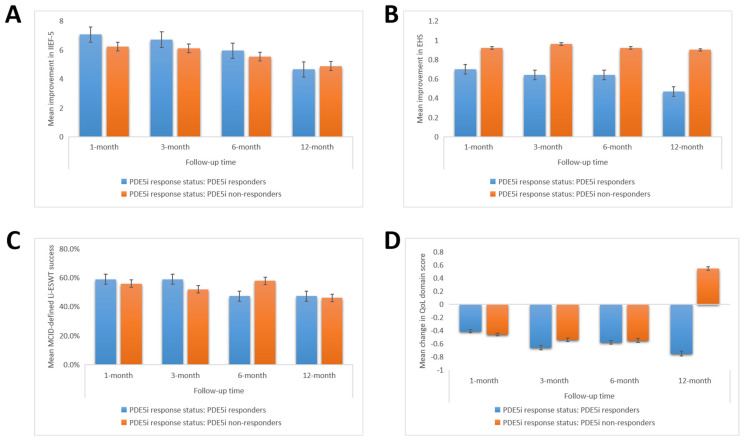
The therapeutic efficacy of Li-ESWT in patients stratified by PDE5i response. Histograms showing the (**A**) mean improvement in IIEF-5, (**B**) mean improvement in EHS, (**C**) mean MCID-defined Li-ESWT success, and (**D**) mean change in QoL domain score in patients with mild, mild-to-moderate, moderate, or severe ED, at indicated time-points. Li-ESWT, low-intensity extracorporeal shockwave therapy; ED, erectile dysfunction; IIEF-5, five-item International Index for Erectile Function; EHS, Erection Hardness Score; MCID, minimal clinically important difference; QoL, quality of life.

**Table 1 biomedicines-09-01670-t001:** Baseline characteristics of the study population (n = 69).

Characteristic	
Age (years, median (Q_1_–Q_3_))	55 (45–66)
Body mass index (kg/m^2^, median (SD))	24.6 (3.89)
Baseline erectile function, median (SD)	
EHS	2.0 (0.80)
IIEF-5	10 (5.54)
ED Severity, n (%)	
Mild (IIEF-5 score: 17–21)	6 (8.6%)
Mild-to-Moderate (IIEF-5 score: 12–16)	16 (23.2%)
Moderate (IIEF-5 score: 8–11)	20 (29%)
Severe (IIEF-5 score: 5–7)	24 (34.8%)
Duration of ED (months, median (Q_1_–Q_3_))	12 (6–36)
PDE5i non-responders, n (%)	52 (75.4%)
Testosterone level (ng/dL, mean (SD))	3.92 (1.45)
Co-morbidity, n (%)	
Diabetes mellitus	29 (42%)
Hypertension	16 (23.2%)
Hyperlipidemia	33 (47.8%)
Chronic kidney disease	9 (13%)
Hypogonadism	21 (30.4%)
Tobacco smoking, n (%)	7 (10.1%)

Q_1_, first quartile; Q_3_, third quartile; SD, standard deviation; ED, erectile dysfunction; EHS, Erectile Hardness Score; IIEF-5, five-item International Index of Erectile Function; PDE5i, phosphodiesterase 5 inhibitor.

**Table 2 biomedicines-09-01670-t002:** Time-phased therapeutic effect of Li-ESWT.

Erectile Function Index	Change from Baseline	
	Mean (95% CI)	*p*-Value
1 month follow-up		
IIEF-5	6.65 (5.30–7.99)	<0.001
EHS	0.88 (0.67–1.09)	<0.001
QoL	−0.45 (−0.74–−0.16)	0.003
Success * n (%)	39 (56.5%)	
3 months follow-up		
IIEF-5	6.37 (5.08–7.67)	<0.001
EHS	0.89 (0.69–1.10)	<0.001
QoL	−0.57 (−0.83–−0.30)	<0.001
Success * n (%)	37 (53.6%)	
6 months follow-up		
IIEF-5	5.79 (4.46–7.12)	<0.001
EHS	0.86 (0.64–1.09)	<0.001
QoL	−0.56(−0.83–−0.29)	<0.001
Success * n (%)	38 (55.1%)	
12 months follow-up		
IIEF-5	5.10 (3.65–6.55)	<0.001
EHS	0.80 (0.58–1.03)	<0.001
QoL	−0.59 (−0.86–−0.32)	<0.001
Success * n (%)	32 (46.4%)	

* Success defined by MCID, minimal clinically important difference; Li-ESWT, low-intensity extracorporeal shockwave therapy; 95% CI, 95% confidence interval; IIEF-5, five-item International Index of Erectile Function; EHS, Erectile Hardness Score; QoL, quality of life.

**Table 3 biomedicines-09-01670-t003:** Negative predictors of Li-ESWT efficacy in our Taiwanese cohort (n = 69).

	Univariate		Multivariate	
	Odds Ratio	*p*-Value	Odds Ratio	*p*-Value
Age (>45 years)	0.27	0.04 *	0.24	0.04 *
BMI (kg/m^2^)	0.90	0.25		
Tobacco smoking	1.30	0.28		
Hypogonadism	0.63	0.40		
Hypertension	0.71	0.54		
Diabetes mellitus	0.43	0.09		
Diabetes mellitus (uncontrolled)	0.17	0.04 *	0.21	0.08
Hyperlipidemia	0.53	0.19		
Hyperlipidemia (uncontrolled)	0.25	0.01 **	0.27	0.03 *
Severe and moderate ED	0.56	0.27		
Duration of ED (>2 year)	0.54	0.38		
EHS < 3	0.62	0.37		
PDE5i non-responders	0.88	0.83		

BMI, body mass index; ED, erectile dysfunction; EHS, Erectile Hardness Score; PDE5i, phosphodiesterase 5 inhibitor; * *p* < 0.05, ** *p* < 0.01.

## Data Availability

The data used in the current study are all contained in the manuscript, and may be obtained upon reasonable request from the corresponding author.

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
