# Peer review of "Efficacy of Penile Low-Intensity Shockwave Therapy and Determinants of Treatment Response in Taiwanese Patients with Erectile Dysfunction"

_biomedicines, 2021, doi:10.3390/biomedicines9111670_

Round 1

Reviewer 1 Report

Dear authors, congratulations for your effort. In general  your work supports the beneficial result of Li-ESWT on erectile dysfunction, indicating that Taiwanese patients with moderate and severe ED benfit the most, in contrast with the current literature.

Please take under consideration some of my remarks upon your work:

  1. First of all your study's plan lack significant components, such as a placebo arm. It's mandatory for any medical research, let alone for such a condition such as ED. Having a single arm deprives the study from substancial data.
  2. In my humble opinion, it seems a little bit dangerous to jump into general conclusions, such as and I quote "the present study showed that Taiwanese patients with ED are mostly older than 45 years and present with moderate and severe ED." Your sample isn't that big so I suggest you avoid such kind of deductions.
  3. You haven't presented solid data supporting your observation that LI-ESWT is more beneficial in moderate and severe ED in Taiwanese patients. What is the underlying mechanism, why these patients benefit the most, in contrast with all the current literature? What makes Taiwanese patients suffering from ED so special?

In general I propose that you reconsider the design of your study. Afterall  you admit it by recognizing the limitations of your study. I know that this is difficult, since it is a retrospective one, but at least you have a good start.

Reviewer 2 Report

Thanks for this well written, enlightening report. Very few corrections would be needed. In results, please delete "interestingly" in the few places where this useless word is used. In discussion, around line 328, is the EU guidance recommending use only in mild cases based on data or is it a cautious effort to minimize possible harm? Please add info here if possible. Around line 334, please break up this overly long run-on sentence.  

Reviewer 3 Report

Kai-Yi Tzou et colleagues presented an interesting article regarding the efficacy of Li-ESWT in a cohort of 69 Taiwanese patients with ED of varying degrees. The topic is absolutely interesting, since the role for Li-ESWT as a treatment for ED is a source of debate by major scientific societies, as also reported by the authors. The results of the study are interesting, as they suggest a significant role for this treatment in improving erectile function. Unfortunately, several critical issues exist in this work.

MAJOR ISSUES

  1. The language needs extensive revision. The writing style, in fact, does not appear scientific and numerous grammatical and punctuation errors emerge that make reading the work difficult.
  2. The authors included only “non-psychogenic ED” cases. It should be stated how the diagnosis of ED was made and how the psychogenic origin of ED was excluded
  3. Statistical analysis: why did not the authors evaluate the normal or non-normal distribution of continuous variables? Why did they choose to report median and DS rather than mean and DS or median and IQR? Why did they use parametric rather than non-parametric tests? Univariate analysis has several limitations, why did not the authors consider a multivariate analysis?
  4. Results: I agree with the choice to report data by differentiating it into paragraphs, but I think it should be better if result were not anticipated by the title of the paragraph, replacing it with the intent of the paragraph (e.g., for the first paragraph, “baseline characteristics of the study population”).
  5. Figure 1: reporting the mean or median values of IIEF, rather than the change, would make reading the graph more intuitive.
  6. Page 8, line 282: there is not any Table 3
  7. Discussion: except as reported in the guidelines, the discussion lacks sufficient comparison with what is the state of the art. Moreover, the authors' judgment seems too unbalanced in favor of Li-ESWTs, whose outcome, in fact, appears modest.

MINOR ISSUES

  1. Page 3, lines 103-109: this part belongs to the “methods” section
  2. Page 3, liens 110-115: this part should not be in the introduction
  3. Page 3, line 116: remove do dot after “methods”
  4. Page 3, line 131: the number of the included patients should be reported just one time
  5. Page 4, line191: the abbreviations DM and HTN have already been used
  6. Please add title + caption for figures
  7. Page 8, line 282: odds ratio has been already reported as OR

Round 2

Reviewer 1 Report

Dear authors, I would like to take advantage of the opportunity that you gave me, answering back to my comments.

Q3: I strongly suggest you to rephrase your words and I quote:"...the present study showed that Taiwanese patients with ED are mostly older
than 45 years and present with moderate and severe ED." to the present study showed that Taiwanese patients with ED, who participated in this study, are mostly older 45 years and present with moderate and severe ED. I believe this rephrasal avoids generalizations.

Q4: I am looking forward for your next work.

Q5: As a retrospective study it is fairly ok not to have a placebo arm, I truly hope that your next study has one.

Reviewer 3 Report

The authors responded positively to the suggestions and, where they disagreed, adequately justified their choices. 

Author Response

We sincerely thank the reviewer for taking time once again to read through our manuscript and evaluate its suitability for publication. We also thank the reviewer for the understanding and professional guidance through provided critiques and suggestions, all of which have helped improve the quality of our work. More so, we are very grateful for the reviewer’s continued interest in our work, and for recommending its acceptance.